# The Influence of Core Self-Evaluations on Group Decision Making Processes: A Laboratory Experiment

**Matteo Cristofaro \*** **, Pier Luigi Giardino and Luna Leoni**

Department of Management and Law, University of Rome 'Tor Vergata', 00133 Roma, Italy;
pierluigi.giardino@alumni.uniroma2.eu (P.L.G.); luna.leoni@uniroma2.it (L.L.)
**\*** Correspondence: matteo.cristofaro@uniroma2.it

**Abstract:** The personal trait called Core Self-Evaluations (CSE) has been receiving increasing attention from behavioral strategy scholars due to its ability to predict job performance and to explain some facets of decision-making processes. However, despite previous studies hypothesizing that managers with high values of CSE are intuitive thinkers, beyond any doubt of their capacities and that they significantly lead to positive results for their organization, no one has empirically investigated these assumptions. This gap can be substantiated by the following research question: *"How do high Core Self-Evaluations influence team decision-making processes?"*. Answering it provides insights on how the evaluations that decision makers make about situations (and the consequent actions that are implemented) highly depend on decision makers' inner traits and their effect on cognition. To fill this gap, 120 graduate students—divided into groups of four—took part in a simulation game and were asked to make decisions acting the role of General Manager of a small-sized manufacturing firm. Tests aimed at identifying the CSE and intuitive/reflecting thinking approach of participants were administered; moreover, the performance resulting from their decision-making processes and their estimation of reached results were collected. Results show that an average level of CSE is preferable to balance intuitive and reflective thinking, as well as avoiding overconfidence bias and reaching the best performance possible. This work suggests that there is a huge misattribution in considering a high level of CSE as being beneficial for decision-making processes and consequent performance.

**Keywords:** behavioral strategy; decision-making; core self-evaluations; intuition; overconfidence; performance

---

## 1. Introduction

In order to survive in the current competitive environment, both private and public organizations are constantly looking to reach organizational effectiveness and efficiency; however, the achievement of these results depends on organizational agents' level of effectiveness and efficiency (Inuwa 2016) and how good they are at making decisions (Simon 1947). According to the established literature, organizational agents' ability to make successful decision-making processes mainly relies, on the one hand, on the organizational environment (Wu and Lee 2016) and, on the other hand, on individual characteristics, such as emotions (Cristofaro 2019, 2020) and personal traits (Paniccia 2002; Cristofaro 2016, 2017a; Busic-Sontic et al. 2017). Among personal traits, the Core Evaluations (CE) (Judge and Bono 2001), namely the evaluations that individuals make about others, the world, and themselves (in this last case, we mean core self-evaluations, CSE), are those that have been receiving increasing attention from behavioral strategy scholars (Hiller and Hambrick 2005; Powell et al. 2011), due to their ability to predict job performance (Judge and Bono 2001) and to explain some facets of decision-making processes (Hollenbeck et al. 1988; Judge et al. 1998; Silvester et al. 2002).

However, despite previous studies hypothesizing that managers with high values of CSE (assimilated to hubris; see Hiller and Hambrick 2005) are intuitive thinkers, beyond any doubt of their capacities and that they significantly lead to positive results for their organization (e.g., Claxton et al. 2015), no one has empirically investigated these assumptions. Studies have investigated some trait pillars comprising CSE, such as self-esteem, in relation to decision-making process variables; however, there has not been any investigation at the group level of analysis (e.g., Jordan et al. 2007), even though the level is a common working unit within organizations (Koontz et al. 1980). Filling this gap, which can be substantiated by the research question: "*how do high Core Self-Evaluations influence team decision-making processes?*", is relevant also from a practical point of view due to the fact that personnel selection managers have been found to be biased and oriented towards selecting people that show high levels of CSE (Cristofaro 2017b).

In order to answer the aforementioned research question, a sample population of 120 graduate students was collected—divided into groups of four—to take part in a simulation game and they were asked to make decisions, in groups, acting the role of General Manager of a small-sized manufacturing firm. Tests aimed at identifying the CSE and intuitive/reflecting thinking approach of participants were administered; moreover, the performance resulting from their decision-making processes and their estimation of reached results were collected. Results of the work show a partial verification of the hypothesis that a high level of CSE leads to an intuitive decision-making process and a high level of performance; meanwhile, they are totally supportive of the positive relationship between a high level of CSE and being victim of overconfidence bias. Yet, results show that the average level of CSE is preferable to balance intuitive and reflective thinking as well as avoiding overconfidence bias and reaching the best performance possible.

Evidence produced really helps to understand the implications of the CSE trait variable for decision-making processes within organizations. Accordingly, prior literature provides evidence of the misattribution in considering a positive value of a high level of CSE as beneficial for performance (Judge et al. 2009; Judge and Hurst 2008). Yet, to the best of the authors' knowledge, this is the first study investigating the CSE trait variable at the group level—apart from also being the first to investigate, in depth, its implications for organizational decision-making processes. These results suggest that, in terms of practical implications, human resource managers track the CSE level of each individual within organizations and suggest composing teams accordingly to achieve better decisional and organizational performance.

The paper is structured as follows. First, readers of *Administrative Sciences* are informed of the basic assumptions behind the Core-Self Evaluation construct and concepts at the basis of decision-making processes. Second, the development of the hypotheses is provided. Third, the methodology is shown with details about the research context, i.e., the simulation game, as well as the data collection and analysis procedures. Fourth, results of the three one-way Analysis of Variance implemented to test developed hypotheses are shown. Fifth and last, discussion of the results in light of prior literature as well as implications for theory and practice are given.

## 2. Theoretical Background

### 2.1. Core-Self Evaluation

Judge et al. (1998) distinguished four components (also called *dispositions*) as being the foremost examined in industrial psychology since the 1960s: (i) Self-Esteem—the general esteem an individual has with respect to himself/herself; (ii) generalized Self-Efficacy—the self-estimation of being successful; (iii) Locus of Control—the conviction in controlling life's variables; and (iv) Emotional Stability—the capacity to maintain a low neuroticism level. These four dispositions have been broadly illustrated as influencing decision-making processes; for example, in job interviews, the self-esteem of candidates was found to be an indicator of work execution (Hollenbeck et al. 1988); this is because candidates with high self-esteem are considered as having control of circumstance (Silvester et al. 2002). Moreover,

Judge and colleagues illustrated that the four highlighted components are altogether inter-related and can be translated as a special degree of core (fundamental) evaluations (CEs) that individuals make about others, the world and themselves (in this final case they are called core self-evaluations; CSE) (Judge et al. 1998; Judge and Bono 2001). In practice, the individual who assesses himself/herself extremely high in these four predispositions is certain of his/her capabilities and is predicted to make successful decision-making processes as well as have positive job and life satisfaction (Judge and Bono 2001).

The CE construction encourages the examination of the 'human factor' within the work environment (Judge et al. 2009). From that, researchers distinguished significant scores in CE as being related to high scores in work interest (Erez and Judge 2001), career accomplishment, objective achievement, wage, and job-related status (Judge and Hurst 2008). In practice, this concept illustrated that individuals who are exceedingly sure in their claim capacities and make a maintained exertion over time towards their objectives (in life and at work), are motivated to achieve effective results in their career by picking up noteworthy compensations and prestigious positions in society (Cristofaro 2017b).

However, management scholars interested in the consideration of the self-concept and the impact on decision-making processes contended that CSE ought to be accepted as a powerful, confirmed umbrella that is created to investigate the executive self-concept. In fact, a really high level of CSE (or hyper-CSE) borders closely with what is regularly, colloquially, called "hubris" (Hiller and Hambrick 2005). In particular, Hiller and Hambrick (2005) investigated the links between executive peculiarities and components of organizational strategy, structure, and execution; from that, they expect that hyper-CSE executives—who have preeminent levels of self-confidence, self-potency, and conviction that they will prevail—will show this characteristic in their work behaviors. For example, they state that managers with high values of CSE are beyond any doubt about their capacities, and they consider profoundly that the application of their capacities will bring positive results. They are free of apprehension and have little concern regarding negative results since they have the conviction that they can overcome difficulties and solve all issues. In short, hyper-CSE executives are sure they will prevail in the work environment as well as in society.

Among the few scholars that empirically investigated CSE in business settings, the work of Nag et al. (2020) is worthy of notice. These scholars are interested in understanding the drivers of SMEs' development whilst competing in declining businesses. In particular, Nag et al. (2020) research looks at how CEO scanning behaviors—investigated in terms of scanning intensity and proactiveness—influence self-efficacy and, in turn, influence firm innovation and performance. Results showed that scanning intensity is emphatically related with self-efficacy; in addition, these scholars reinforced the idea for an arbitrative relationship in which CEO self-efficacy moderates the impact of scanning intensity on SME execution and development.

Finally, similarly to the CSE, Zell et al. (2020) introduced the concept of BTAE (Better-Than-Average-Effect), conceptualized as the propensity of individuals to see their capacities, traits, and identity characteristics as predominant compared with their normal peer. They displayed an exhaustive meta-analysis of BTAE investigation, collecting information from 124 scientific empirical articles, including 291 hypothesis tests on more than 950,000 people. Results advised that the BTAE is related with self-esteem and life fulfillment—so, hypothesizing an overlap between CSE and BTAE.

### 2.2. System Thinking, Biases and Overconfidence

'Biases' are deviations from rational choice, and this umbrella term has been used over time for substantiating both cognitive traps (i.e., mental errors) and heuristics (i.e., cognitive shortcuts) (Cristofaro 2017a, 2018). The first group of biases has been initially and formally defined by Hammond et al. (1998), inspecting cognitive errors behind executive choices. The second specifically comes from the program, primarily conducted through laboratory experiments of Kahneman and Tversky in the 70s on heuristics: rules that are deliberately, or not, used when choice conditions are dubious or complex (due to the massive or scarce amount of information)

(Kahneman and Tversky 1974, 1979). However, these shortcuts were expected to be connected with the functioning of the human mind. In fact, Kahneman (2003)—the main theorist of the so-called *dual process theory*—described human cognition as working according to two diverse Systems of our intellect: System 1, deputed to cognitive activities that are quick and programmed, and System 2, committed to cognitive activities that are "consciously observed and purposely controlled" (Kahneman 2003, p. 698). From this explanation, System 1 is the primary to be stimulated during our day-by-day tasks; so, human reasoning depends on heuristics for most of our mental procedures (Kahneman 2011).

However, traps and heuristics bring distinctive results on decision making: continuously negative for traps and positive or negative for heuristics. Undoubtedly, despite heuristics having been firstly conceived as the second-best choice approach (Kahneman et al. 2011), the *ecological rationality* aspect of heuristics has assumed and demonstrated as bringing, under some conditions, better choices than would be the case if decision makers utilized one of the more complicated strategies to make a choice (e.g., logistic regression; Luan et al. 2019).

Among traps, one of the most studied has been the overconfidence trap, defined by Bazerman and Moore (2013) as the "mother of all biases", since it activates other biases with dramatic consequences on the entire decision-making process (Abatecola et al. 2018). Overconfidence is the circumstance in which people tend to be overbearing regarding the accuracy of their judgments and it has been found in numerous settings (Cain et al. 2015; Gudmundsson and Lechner 2013). For example, Shiller (2005) illustrates that the stock markets were overestimated both in the case of the dot-com and real estate bubbles, primarily due to the executives' overconfidence in getting large returns, which progressively pushed to bring back and collect only corroborative data. Yet, Chen et al. (2015) examined how overconfidence may happen when executives make a corporate's profit projections. Examining the contrast between, at least, two large companies' earning estimations, made by 217 CEOs in a 14-year time span (1994–2008), Chen and colleagues identified significant steady outcomes about their assumption of the fact that CEOs with more prominent overconfidence are safe against strong remedial criticism. However, as Park et al. (2011) clarified, CEO overconfidence may be crucially decided by high levels of adulation, in terms of approbation and reaction similarity (or affirmation) from other members of the board of directors. These scholars explored this aspect through investigating 451 CEOs of US corporations and 3135 other executives. The outcomes completely bolster the initial theories that CEOs with a high social status are emphatically related with CEO overconfidence; this give-and-take brings a slight recognition of the need to modify procedures in return for mediocre conduct.

Studies on overconfidence have been massive and they have not followed, as demonstrated, the same conceptualization and operationalization of overconfidence. In this vein, Moore and Healy (2008) synthesized these different approaches within three clusters of how overconfidence has been investigated: (i) overestimation of one's substantial conduct (the majority of studies adopted this definition), (ii) overplacement of one's conduct relative to others, and (iii) extreme accuracy in one's convictions. Empirical evidence displays that the inversions of the first two (apparent underconfidence) tend to differently influence organizational tasks. On the one hand, for severe tasks, individuals overestimate their real performance, but also erroneously accept that they are less effective than others; on the other hand, for simple assignments, individuals underestimate their real performance but erroneously believe they are better than others.

From the aforesaid, overconfidence has a great impact on management decisions and, because of that, scholars have tried to explain whether there are some personal and/or contextual variables that may foster or reduce it. In this regard, Croskerry and Norman (2008) found, for example, that overconfidence is related with evidence favoritism—thus, the confirmation bias. In their study of clinical decision making, overconfidence has been generally found acting as deferring and/or missing necessary analysis, highlighting the catastrophic impacts of this decision behavior. In addition, Mata et al. (2013) studied whether the thinking mode—deliberative versus intuitive—that individuals utilize to figure out a dilemma or make a judgment impacts the mindfulness of decision makers' claims and others' conduct. Findings show that contemplative thinkers had a metacognitive superiority over

instinctive thinkers. Deliberative individuals are aware of both the deliberative arrangement and the instinctive alternative; realizing that the deliberative arrangement is superior, they are likely to feel surer and be more precise in how they evaluate their execution. Instinctive thinkers, on the other hand, are aware only about the instinctive explanation; they know only this option, so they are unconcerned of how unaware they were of their conduct and how they rank in relation to others.

## 3. Hypotheses' Development

Hayward and Hambrick (1997) used intermediary indicators of hubris, which captured a variety of presumed circumstantial and personality aspects, demonstrating the crucial need for a psychometrically grounded and approved construct for studying the extraordinary self-confidence in executives. Later, Hiller and Hambrick (2005) considered hyper-CSE to precisely measure that construct; the upper limit of CSE may be considered as an accurately certified 'hubris factor'. In sum, analysis on the conceptually comparable ideas of executive narcissism and hubris led our understanding of high-CSE executives. Especially, a significant level of CSE may precisely coincide to what is colloquially pointed out as hubris. Hubris, or hyper-CSE, has a great connection with intuitive thinking, as advanced by Claxton et al. (2015) who claimed that hubristic leadership is usually connected with intuition.

Jordan et al. (2007) examined both intuition and the correlation between tacit and explicit self-esteem, one of the four fundamentals of CE, testing whether the grasped effectiveness of the intuition increases the congruity between tacit and explicit self-esteem. It appeared that individuals who persistently consider their instinct as predominant are more inclined to tacit and explicit self-esteem. Contrarily, individuals with moderately intuitive thinking inclination had a negative correlation between tacit and explicit self-esteem, proposing that they may overcorrect their explicit self-views. Translating the above mentioned at the collective level:

**Hypothesis 1.** *Teams with high CSE are more intuitive compared to teams with low and average CSE.*

Kramer et al. (1993) explored the influence of motivational and emotional mechanism on negotiator judgment. They considered whether positive disposition and the inspiration to preserve high self-esteem lead the negotiator to be overconfident and to make excessively positive self-evaluation. A research test using dyadic bargaining was organized to test this hypothesis and outcomes supported Kramer's forecasts that high self-esteem and positive attitude influenced negotiators' determination and confidence preceding to negotiations, as well as their post-negotiation assessments of conduct. Similarly, Baumeister et al. (1993) analyzed the inclination for individuals with high self-esteem to judge themselves about their ability to make commitments; it resulted that subjects with a high level of self-esteem end up setting inappropriate, risky goals that were beyond their capabilities, so they finished with smaller rewards than subjects with low self-esteem. Yet, Zacharakis and Shepherd (2001) analyzed whether Venture Capitalists (VCs) are victims of overconfidence when evaluating firms' potential, as well as the elements surrounding the choice that lead to overconfidence. The outcomes of their experiment demonstrated that VCs are undoubtedly overconfident (96% of the 51 VCs showed a critical level of overconfidence), which negatively influences VC's decision efficiency. In particular, they found that when VCs are familiar with decision-making processes, such as the evaluation of venture success, and the structure of the data included that choice, they resort to automatic information processing; they rely on limited information, leading them to fall victim to overconfidence. This has also been developed by a few later articles on the relationship of General Managers' dispositions and their capacity to carry on sustainability practices; specifically, Abatecola and Cristofaro (2019) demonstrated, through a literature review, that CEOs with large CSE are mostly certain in their analysis in carrying out unsustainable business practices. Translating the above mentioned at the collective level:

**Hypothesis 2.** *Teams with high CSE are more inclined to be victims of the overconfidence bias compared to teams with low and average CSE.*

As already mentioned, Hiller and Hambrick (2005) found that a high level of CSE is close with what is routinely called "hubris". They have also explored the connections between executive distinctiveness and factors of organizational technique, structure, and execution; from that, they anticipate that hyper-CSE managers will display this characteristic in their work attitude. They state that executives with large CSE are beyond any questioning of their skills, and they consider significantly that the application of their competences will bring positive outcomes. CEs outlined that people who are extraordinarily beyond any doubt in their claim abilities, and make continued effort over time towards their targets, are persuaded to get compelling results in their career by picking up essential compensations and important roles in society. Yet, the link between CSE and performance has also been proved to work when judging candidates for job vacancies. In particular, Cristofaro (2017b) found, through a laboratory experiment involving personnel selection executives, that candidates who are perceived to have high CSE are also those that will achieve great performance; in contrast to low CSE candidates that are perceived as low performers. Translating the above mentioned at the collective level:

**Hypothesis 3.** *Teams with high CSE reach higher positive performance compared to teams with low and average CSE.*

## 4. Methodology

### 4.1. Research Context

For several decades, scholars used business games to investigate the accuracy of managerial decisions and their application through an analysis of the performances made by the participants of their experiments (Lainema and Makkonen 2003; Faria et al. 2009; Kim et al. 2013; Henriksen and Børgesen 2016; Korchinskaya et al. 2020). In this section, the simulation game 'General Management Business Game (GMBG)' (provided by Artémat) is briefly explained, which was used during the lectures of a general management course in one large Italian University, for training students in the management of firms as well as for doing this research. In this simulation game, students take the part of a General Manager of a small-sized manufacturing firm specialized in the production of mobile devices (e.g., Tablet.). Players aim to win contracts to manufacture and sell a large selection of articles, while tuning their businesses through the benchmarks of operations governance. This developing organization requires participants to take care of both the human and productive resources tailoring the optimization capacity, and because of this they should make several management decisions that touch all the aspects of the organization. This is proved within a module of the business game in which participants are asked to run all the managerial functions of the firm with the aim of achieving the highest possible performance for the firm—measured in terms of net worth, number of contracts signed, and gratification of clients. Students cross paths through "real-job" alternatives allowing them to work on management decision-making processes by collecting and analyzing data. The main performance indicator used to derive how the simulated firm was conducted is the net worth: the value of all the non-financial and financial assets owned by the firm minus the value of all its outstanding liabilities.

### 4.2. Experimental Design, Procedure and Measurements of Variables

To test the three developed hypotheses and answer the research paper's aim, 120 graduate students (57 male, 63 female, Average Age = 21.2 years, Standard Deviation (SD) = 1.3 years) following an optional management course in one large Italian University were involved in this research; participants, that can be considered as 'Millennials'; participants were rewarded for the experiment with University credits. The laboratory experiment, then followed by quantitative analysis, is considered as the most suitable research design in research fields that can be considered at a stage of development far from the nascent one (Edmondson and McManus 2007). These intermediate or mature fields of research—like

the role of personal traits in decision making processes (see the review by Cristofaro 2017a)—are challenged by "focused questions and/or hypothesis relating existing constructs" (p. 1160).

Sampled students, as gleaned from informal conversation, had no or limited work experience; however, we did not control work experience in an empirical way. The selection of participants followed the convenience sampling approach (Given 2008), which is non-probability sampling that consists of selecting a sample from a part of the population that is close at hand. Despite the fact that a stream of scholars believe that students' samples are not suitable for behavioral research aiming to provide working implications for practitioners (e.g., Gallander Wintre et al. 2001), another equally important stream of research (e.g., Lucas 2003; Thomas 2011) believes that these samples are appropriate in cases of research emphasis on basic psychological processes. Concerning the latter, according to Berkowitz and Donnerstein (1982): "the meaning the subjects assign to the situation they are in and the behavior they are carrying out plays a greater part in determining the generalizability of an experiment's outcome than does the sample's demographic representativeness" (p. 249). So, here it is strongly believed that due to the aim of this research (thus, finding connections among CSE, overconfidence and decision-making performance) and the settings of the laboratory experiment, at least the internal validity of the research is guaranteed. This follows similar contributions on management decisions (see Cristofaro 2016) that consider students' samples and investigate psychological variables against decision-making performance.

At the beginning of the course, each participant was free to form a group with three other members; a total of 30 groups were composed. The experiment took place when participants were asked to run the final module of the simulation game in which they had all the managerial functions under control with the aim of grasping the most elevated conceivable execution for the firm. On the day of the experiment (when students were asked to play this final module of the simulation game), the leading researcher explained the interest in studying the relationship between their personality traits, some behavioral decision variables and their connections with performance in the simulation game.

Before running the final module of the simulation game, each participant was first invited to answer the 12-item CSE Scale (CSES) (Appendix A) and the seven-item Cognitive Reflection Test (CRT) (Appendix B) to measure their CSE degree and tendency to adopt intuitive or reflective thinking (explained later). Then they were asked to perform the last module of the simulation game; this occurred over 3 hours. During this time period, participants were not aware of their performance (in terms of net worth; tracked by the leading researcher). At the end of the simulation game, each individual, within groups, was asked to estimate the final net worth of their firm and to note it on a paper. Variables at the center of the developed hypotheses to be tested were measured as follows.

*Core Self-Evaluations:* The CSE score of each group was calculated according to the average of the CSE score of its members. The CSE score at the individual level was derived by asking participants to complete the 12-item Core Self-Evaluations Scale (CSE by (Judge et al. 2003) in Appendix A; test-retest accuracy was 0.81 over a one-month span) which, rather than measuring the four characteristics comprised within the CSE independently and weighting the scores, provides an explicit and integrative estimation of a person's core self-perception. Respondents were asked to rank their predisposition on the 12 items according to a five-point Likert scale ranging from 1 (strongly disagree) to 5 (strongly agree) (see also Joo et al. 2012). It is worth noticing that values assigned to reverse grade questions were subtracted; from that, the maximum and minimum value that can be reached by completing the CSE are, respectively, +24 and −24, with a neutral point at 0. Thanks to the STATA function called 'egen'[1], it was possible to derive three main clusters according to the average CSE value of groups: (i) low CSE groups (whose averages range from −24 to −9), (ii) average CSE groups (whose averages range from −8 to +8), and (iii) high CSE groups (whose averages range from +9 to +24). Thanks to that, the initial 30 groups were reallocated in these three clusters, equating to 10 groups each.

---

[1]   computed as: egen CSE = cut(CSE), at(−9,8,24).

*Reflective and intuitive thinking:* To assess the cognitive predisposition of respondents between reflective and intuitive thinking, the seven-item CRT by Toplak et al. (2014) was adopted (Appendix B), which extends the three-item CRT formulated by Frederick (2005). The seven questions are constructed such that they have an instinctive but erroneous reply that arises rapidly and an appropriate reply that is simple to catch when it is clarified. Subsequently, the test is presumed to estimate a person's tendency to engage in intuitive or reflective thinking (Patel et al. 2019). Each correct answer was counted as 1, while an incorrect answer was 0; so, the maximum and minimum values that can be reached by completing the CRT are, respectively, 7 and 0. After having calculated the sum of correct and incorrect answers for each individual, this value was summed with the ones of the other members of the group and a final average was derived. Among the participants who did not yield the proper reply, the instinctive one was usually the most given answer.

*Overconfidence* (overestimation): Moore and Healy (2008), as already introduced, defined that overconfidence usually has been measured in terms of overestimation and this is the definition of overconfidence that has been operationalized for this study. In particular, during the simulation game participants were not made aware of their current net worth; at the end of their performance, each individual, within groups, was asked to make an estimation of their final net worth. After collecting each individual's estimation of their net worth, this value was summed with those of the other members of the group and a final average was derived. To calculate the overestimation of each group, the final average of the estimation of the net worth of the group was compared with the actual net worth and a subtraction was made to find the overall overestimation of their results. This procedure is in line with the one adopted by Hoppe and Kusterer (2011).

*Performance*. To measure the groups' performance—the outcome of management decisions that have been collectively made by groups—during the simulation game, the net worth variable (as suggested by McGraw Hill's instruction material) was taken as the best representative of performance reached within the simulation game. The net worth is the value of all the non-financial and financial assets owned by the firm minus the value of all its outstanding liabilities; this is automatically calculated by the software. This variable has also been used as an indicator of firms' performance by other authors in the past (Penrose 1956; Carlstrom and Fuerst 1997).

*Data analysis*: To test the three developed hypotheses, three one-way Analysis of Variance (ANOVA) have been implemented. This is the most suitable statistical technique that can be used in order to compare means of two or more samples to find significant differences, if any (Field 2013). A Tukey post hoc test was conducted after each One-way ANOVA to determine the significant differences among groups.

## 5. Results

To verify whether teams with high CSE are more inclined to intuitive thinking rather than average and low CSE groups (H1), a one-way ANOVA was firstly implemented considering the different CSE clusters (low, average, high) and their groups' results on the CRT test.

As shown in Table 1, there was a statistically significant difference between groups as determined by the one-way ANOVA ($F_{(2,27)} = 80.510$, $p = 0.000$). A Tukey post hoc test, shown in Table 2, revealed that high CSE groups were more inclined to the average CSE groups for intuitive thinking (−4.2 right answers compared with them; $p = 0.00$), but they were equally inclined to intuitive thinking with respect to low CSE groups ($p = 0.964$) ($p = 0.989$). So, H1 is *partly* verified.

**Table 1.** ANOVA Table—CSE on intuitive-reflective thinking.

|  | Sum of Squares | df | Mean Square | F | Sig. |
|---|---|---|---|---|---|
| Between Groups | 120.467 | 2 | 60.233 | 80.510 | 0.000 |
| Within Groups | 20.200 | 27 | 0.748 |  |  |
| Total | 140.667 | 29 |  |  |  |

**Table 2.** Post-hoc test—CSE on intuitive-reflective thinking.

| (I) CSE | (J) CSE | Mean Difference (I-J) | Std. Error | Sig. | 95% Confidence Interval | |
| --- | --- | --- | --- | --- | --- | --- |
| | | | | | Lower Bound | Upper Bound |
| Low CSE | Average CSE | −4.30000 * | 0.38682 | 0.000 | −44.181 | −41.819 |
| | High CSE | −0.10000 | 0.38682 | 0.964 | −0.2181 | 0.0181 |
| Average CSE | Low CSE | 4.30000 * | 0.38682 | 0.000 | 41.819 | 44.181 |
| | High CSE | 4.20000 * | 0.38682 | 0.000 | 40.819 | 43.181 |
| High CSE | Low CSE | 0.10000 | 0.38682 | 0.964 | −0.0181 | 0.2181 |
| | Average CSE | −4.20000 * | 0.38682 | 0.000 | −43.181 | −40.819 |

* The mean difference is significant at the 0.05 level.

In order to verify whether teams with high CSE are more inclined to be victims of the overconfidence bias compared to teams with low and average CSE (H2), a one-way ANOVA was implemented considering the different CSE clusters (low, average, high) and their average teams' estimation of performance.

As shown in Table 3, there was a statistically significant difference between groups as determined by the one-way ANOVA (F(2,27) = 295.962, $p$ = 0.000). A Tukey post hoc test, shown in Table 4, revealed that high CSE groups were more inclined to the average CSE groups (+\$12,000 of net worth overestimation compared with them; $p$ = 0.00) and low CSE groups (+\$30,000 of net worth overestimation compared with them; $p$ = 0.00) to be victims of the overconfidence bias. So, high CSE groups were the ones that overestimated their performance more than other CSE groups, while low CSE groups were the ones that overestimated their performance least compared to other CSE groups. So, H2 is verified.

**Table 3.** ANOVA Table—CSE on overconfidence.

| | Sum of Squares | df | Mean Square | F | Sig. |
| --- | --- | --- | --- | --- | --- |
| Between Groups | 4.560 | 2 | 2.280 | 295.962 | 0.000 |
| Within Groups | 208,000,000 | 27 | 7,703,703.70 | | |
| Total | 4.768 | 29 | | | |

**Table 4.** Post-hoc test—CSE on overconfidence.

| (I) CSE | (J) CSE | Mean Difference (I-J) | Std. Error | Sig. | 95% Confidence Interval | |
| --- | --- | --- | --- | --- | --- | --- |
| | | | | | Lower Bound | Upper Bound |
| Low CSE | Average CSE | −18,000.00 * | 1241.26 | 0.000 | −18,379.008 | −17,620.992 |
| | High CSE | −30,000.00 * | 1241.26 | 0.000 | −30,379.008 | −29,620.992 |
| Average CSE | Low CSE | −18,000.00 * | 1241.26 | 0.000 | 17,620.992 | 18,379.0077 |
| | High CSE | −12,000.00 * | 1241.26 | 0.000 | −12,379.008 | −11,620.9923 |
| High CSE | Low CSE | −30,000.00 * | 1241.26 | 0.000 | 29,620.9923 | 30,379.0077 |
| | Average CSE | −12,000.00 * | 1241.26 | 0.000 | 11,620.9923 | 12,379.0077 |

* The mean difference is significant at the 0.05 level.

In order to verify whether teams with high CSE reach higher positive performance compared to teams with low and average CSE (H3), a one-way ANOVA was implemented (see Table 5) considering the different CSE clusters (low, average, high) and their average teams' actual performance in terms of net worth.

**Table 5.** ANOVA Table—CSE on actual performance.

|  | Sum of Squares | df | Mean Square | F | Sig. |
|---|---|---|---|---|---|
| Between Groups | 5.388 | 2 | 2.694 | 113.384 | 0.000 |
| Within Groups | 6.415 | 27 | 2,376,000,000 |  |  |
| Total | 6.030 | 29 |  |  |  |

As shown in Table 5, there was a statistically significant difference between groups as determined by the one-way ANOVA (F(2,27) = 113.384, *p* = 0.000). A Tukey post hoc test, shown in Table 6, revealed that high CSE groups reached greater performance than low CSE groups (+\$50,800 of net worth overestimation compared with them; *p* = 0.00), but they reached lower performance than average CSE groups (who, on average, gained +\$53,000 of net worth). So, H3 is partly verified.

**Table 6.** Post-hoc test—CSE on actual performance.

| (I) CSE | (J) CSE | Mean Difference (I-J) | Std. Error | Sig. | 95% Confidence Interval | |
|---|---|---|---|---|---|---|
|  |  |  |  |  | Lower Bound | Upper Bound |
| Low CSE | Average CSE | −103,800.00 * | 6893.47518 | 0.000 | −105,904.8518 | −101,695.1482 |
|  | High CSE | −50,800.00 * | 6893.47518 | 0.000 | −52,904.8518 | −48,695.1482 |
| Average CSE | Low CSE | 103,800.00 * | 6893.47518 | 0.000 | 101,695.1482 | 105,904.8518 |
|  | High CSE | 53,000.000 * | 6893.47518 | 0.000 | 50,895.1482 | 55,104.8518 |
| High CSE | Low CSE | 50,800.000 * | 6893.47518 | 0.000 | 48,695.1482 | 52,904.518 |
|  | Average CSE | −53,000.00 * | 6893.47518 | 0.000 | −55,104.8518 | −50,895.1482 |

* The mean difference is significant at the 0.05 level.

## 6. Discussion and Implications

This work offers a contribution to the debate about how CSE influences team decision-making processes. In order to do so, the research has been based on the analysis of four variables: CSE, intuitive/reflective thinking, overconfidence, and performance. Accordingly, three hypotheses have been formulated and tested—through one-way ANOVA and Tukey post hoc tests—on a sample population composed of 120 students while taking part in a simulation game in which they were asked to make decisions, in groups, acting in the role of the General Manager of a small-sized manufacturing firm.

The results only partially verified the first hypothesis. In fact, both teams with high and low CSE are equally inclined to intuitive thinking. This aspect deserves particular attention; indeed, if it is true that previous literature on the topic (e.g., Hiller and Hambrick 2005; Jordan et al. 2007; Claxton et al. 2015) has already recognized the existence of a link between high levels of CSE and intuitive thinking, it is surprising to see that also groups with a low level of CSE have the same predisposition to intuitive thinking. Therefore, the high self-consideration by individual/groups seems to lead to the same consequence of having low self-consideration: being inclined to intuitive thinking. Despite that, this result can be considered in line with the study of Rudolph et al. (2009) (see also the similar one by Cristofaro 2016) who found, through a computer-based simulation on data collected in clinical decision making, that the decision-making behavior that usually leads to wrong decision options is usually carried out by people that take too little, or too much, time to make a decision. These results complete these studies by providing the explanation at a personal trait level of why this different decision behavior occurs. The results totally confirm the second hypothesis. In fact, teams with a high CSE level were more predisposed to both average and low CSE groups of falling into the overconfidence trap. This is perfectly in line with previous literature on these topics (e.g., Baumeister et al. 1993; Kramer et al. 1993; Zacharakis and Shepherd 2001) and highlights how high CSE groups, overestimating

their capabilities, tend to "destroy" their decision-making ability (Abatecola et al. 2018; Abatecola and Cristofaro 2019). Lastly, results only partially verified the third hypothesis. In fact, if it is true that high CSE groups reached higher positive performance compared to low CSE groups—as already demonstrated by previous researches (e.g., Hiller and Hambrick 2005; Cristofaro 2017b); it is also (surprisingly) true that average CSE groups reached more positive performance compared to high CSE groups. By linking the results of the test of the first and third hypothesis, it emerges that groups with a high level of intuitive thinking (corresponding to the ones having high or low CSE scores) are not the best performers in decision-making terms. This result contributes to the debate on the consequences of intuition in management decision making. In particular, it supports a stream of prior results highlighting that intuitive thinking leads to poor quality of decisions (Elbanna et al. 2013), which consequently leads to poor firm performance (Goll and Rashe 1997). This happens, as suggested by Elbanna et al. (2013), because intuitive decision makers are impatient with routine or details—i.e., they have a poor systemic search for information—and are pushed, by their nature, to quickly reach conclusions and to ignore negative problems. However, despite reinforcing this stream of works, another important one has found a positive relationship between intuition and firm performance; such as in developing technologies, sizing new opportunities, and providing effective responses to crises (Bullini Orlandi and Pierce 2020). What is the determinant for the success of intuitive thinking seems to be, according to Bullini Orlandi and Pierce (2020), the dynamicity of the industry environment; indeed, in cases of highly dynamic and turbulent environments with the presence of real-time data, intuitive thinking is preferred rather than the reflective one. In sum, despite the confirming results of this work in substantiating a negative role of intuitive thinking in decision making, the rapid change of contextual and environmental variables can lead to positive effects of intuitive thinking—in line with the ecological rationality approach (Gigerenzer and Brighton 2009).

Thanks to this work, the results provided extend those of cited and discussed contributions, offering a more solid base for the highlighted assumptions by providing an empirical assessment of an established personal trait variable, CSE. Indeed, cited studies only assumed this relationship looking at one of the four pillars of the CSE, such as self-esteem (e.g., Kramer et al. 1993; Baumeister et al. 1993; Jordan et al. 2007), or by providing a theoretical explanation (Hiller and Hambrick 2005; Abatecola et al. 2018; Abatecola and Cristofaro 2019). Moreover, this is the first study that investigates the outlined relationship at a group level, practically overcoming the limits of the others that considered only the individual level of analysis.

Based on the exposed results, some important managerial implications can be derived for practitioners, especially the younger ones (Millennials) with a similar age to the sampled students. Firstly, as CSE is a personal trait, it is not possible to suppress it in an absolute sense; or, at least, it is very difficult in a short or medium range timescale. However, practitioners can reduce its value to an average by composing an ad hoc team. They can, in practice, bring together people with different CSE levels so that the CSE average will result as "acceptable"—namely, the score of their CSE needs to be between −8 to +8 points—there will be a balance between intuitive and reflective thinking in the team and, thus, the possibility of avoiding the overconfidence trap and the opportunity of achieving satisfactory performance. Thus, it is fundamental that human resource managers track the CSE level of each individual within organizations. In this way, they also have the possibility of appropriately suggesting—to department or unit heads—the "best team composition" to achieve better decisional and organizational performance. At the same time, if an organization needs to quickly respond to internal or external pressures, such as identifying a commercial strategy to counteract a sudden, huge price cut of a competitor, composing teams of individuals with exclusively high or low CSE levels can be beneficial for the production of intuitive (and quick) responses.

Despite the rigor with which the experiment was conducted, this study has some limitations, which also represent fruitful starting points for future researches on these topics. Firstly, the sample population is composed by students, which means that most have little or no work experience. A second and connected limit is determined by the fact that the experiment was conducted during

course lectures; therefore, participants applied their strategies in an environment in which they were comfortable. In particular, they acted without being subjected to external pressure and this could have biased, for example, their risk orientation—with obvious consequences on the potential performance that would have been attained with another 'purer behavior'. A third and no less important limitation of this research arises from the fact that some variables have not been controlled, for example, work experience and risk orientation, although in real life they obviously affect the behavior and attitude of managers in managing situations and acting upon them. Finally, a fourth and last research limit results from the structure of the platform provided to the students. Indeed, like all simulation games, there are limitations about the representation of all the variables that are played within real world choices. In sum, despite the fact that respondents' decisions have been implemented (and tracked by the researchers) within a very well simulated environment (i.e., the simulation game), which is better than paper-based cases, sampling managers in real life situations would be necessary for extending the generalizability of these results. Future research can surely solidify the results of this work by avoiding the outlined limits; moreover, following Abatecola et al. (2018), future research avenues could investigate if there are other cognitive distortions (e.g., self-serving, emotion and cognition collision) in managerial decision making—in addition to overconfidence—that are linked to and influenced by CSE. Yet, the results of this work should be highly considered by scholars that want to deepen the antecedents of intuition and its outcomes in strategic decision-making (see Elbanna et al. 2013). In particular, future research can test whether trait variables—CSE above all—have more weight than contextual and environmental ones in determining the thinking style of decision makers. Last but not least, the link between CSE and performance could be deepened also by looking at the emotional answers that high, average, and low CSE groups have when facing some decisional situations. These can reinforce the debate and operationalization of intuitive thinking which, nowadays, still does not take the role of emotions in substantiating intuitive answers into very high consideration.

The originality of this work is threefold. Firstly, we are not aware of any study that has investigated the influence exercised on decision-making processes by CSE in relation to reflective/intuitive thinking, overconfidence, and performance. Secondly, these relationships have always been investigated at the "individual level". However, most of the tasks in an organization are performed at team level and decisions are rarely made individually; thus, this study adopts a "team/group level" to evaluate the importance and the influence of these aspects on organizations. Thirdly and lastly—despite the limitations previously exposed—this paper gives evidence of what really happens within organizations without resorting to case studies but, instead, is shown through the direct participation of individuals—i.e., students—acting as General Managers.

**Author Contributions:** Conceptualization, M.C.; methodology, M.C. and P.L.G.; formal analysis, M.C.; data curation, M.C.; writing—original draft preparation, M.C. and P.L.G.; writing—review and editing, L.L.; visualization, M.C.; supervision, M.C. All authors have read and agreed to the published version of the manuscript.

**Funding:** This research received no external funding.

**Conflicts of Interest:** The authors declare no conflict of interest.

## Appendix A  Core-Self Evaluation Scale

1.  _____ I am confident I get the success I deserve in life.
2.  _____ Sometimes I feel depressed. (reverse grade)
3.  _____ When I try, I generally succeed.
4.  _____ Sometimes when I fail, I feel worthless. (reverse grade)
5.  _____ I complete tasks successfully.
6.  _____ Sometimes I do not feel in control of my work. (reverse grade)
7.  _____ Overall, I am satisfied with myself.
8.  _____ I am filled with doubts about my competence. (reverse grade)
9.  _____ I determine what will happen in my life.

10.　_____ I do not feel in control of my success in my career. (reverse grade)
11.　_____ I am capable of coping with most of my problems.
12.　_____ There are times when things look pretty bleak and hopeless to me. (reverse grade)

### Appendix B  Seven-Item Cognitive Reflection Test

1.  A bat and a ball together cost 110 cents. The bat costs 100 cents more than the ball. How much does the ball cost? (intuitive answer: 10 cents; reflective answer: 5 cents).
2.  If it takes 5 machines 5 min to make 5 widgets, how long would it take 100 machines to make 100 widgets? (intuitive answer: 100 min; reflective answer: 5 min).
3.  In a lake, there is a patch of lily pads. Every day, the patch doubles in size. If it takes 48 days for the patch to cover the entire lake, how long would it take for the patch to cover half of the lake? (intuitive answer: 24 days; reflective answer: 47 days).
4.  If John can drink one barrel of water in 6 days, and Mary can drink one barrel of water in 12 days, how long would it take them to drink one barrel of water together? (intuitive answer: 9; reflective answer: 4).
5.  Jerry received both the 15th highest and the 15th lowest mark in the class. How many students are in the class? (intuitive answer: 30; reflective answer: 29).
6.  A man buys a pig for £60, sells it for £70, buys it back for £80, and sells it finally for £90. How much has he made? (intuitive answer: £10; reflective answer: £20).
7.  Simon decided to invest £8000 in the stock market one day early in 2008. Six months after he invested, on July 17, the stocks he had purchased were down 50%. Fortunately for Simon, from July 17 to October 17, the stocks he had purchased went up 75%. At this point, Simon has: a. broken even in the stock market. b. is ahead of where he began. c. has lost money. (intuitive answer: b; reflective answer: c value is £7000).

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
