# Peer review of "The Influence of Core Self-Evaluations on Group Decision Making Processes: A Laboratory Experiment"

_admsci, doi:10.3390/admsci10020029_

Round 1

Reviewer 1 Report

The paper has been authored with great professionalism. The research goal, the formulation of the hypotheses and the presentation of the results have been developed with outstanding clarity and details. The discussion, in particular, is very insightful and presents interesting implications and "food for thoughts" for scholars and pratictioners alike. From a formal standpoint, I report a clear presentation if the ideas without spelling mistakes or typos.

Author Response

Ms. Ref. No.: admsci-784794

The Influence of Core Self-Evaluations on Group Decision Making Processes: A Laboratory Experiment

for

Administrative Sciences

General reply to reviewers

Dear Editor and Reviewers,

We would like to express our gratitude for your comments in this round of review. We thank the editors and reviewers for the guidance to refine our work.

Please find attached the revised version of our manuscript entitled “The Influence of Core Self-Evaluations on Group Decision Making Processes: A Laboratory Experiment”.

Thanks for reading and taking into consideration our manuscript. Your comments were well-received and the relative suggestions have been incorporated into this improved version. Your work truly enables us to significantly improve our own.

In the following section, we have reported the comments and our replies. In the text, all of the changes are highlighted in yellow.

***

Reviewer: 1

Recommendation: Accept

General comment

The paper has been authored with great professionalism. The research goal, the formulation of the hypotheses and the presentation of the results have been developed with outstanding clarity and details. The discussion, in particular, is very insightful and presents interesting implications and "food for thoughts" for scholars and practitioners alike. From a formal standpoint, I report a clear presentation if the ideas without spelling mistakes or typos.

Reply to general comment: Thank you for having appreciated the contribution, its aim, organization and development! Your immediate acceptance response is truly appreciated; this is a very great recognition for a scientific work and scholars behind it. Thank you.

Yet, thank you for the comment on improving the communication style. We have asked Mrs. Alex Kessler, who has been a professional proofreader of academic works for the last 10 years and that is a native speaker of English, to assist us in polishing the work.

Yours sincerely,

The Authors

Reviewer 2 Report

Dear Authors,

Thank you very much for giving me the chance to read and review your paper about Core-Self Evaluation (CSE) and its relationships with intuitive/analytical information processing and decision-making performance. The topic is relevant and interesting in the actual debate.

The paper is really well-developed and written; I have only a couple of comments I hope can be helpful to improve the contribution of the paper.

Probably one of the main weaknesses of paper is the employment of graduate students as a unique source of data; nevertheless the rigor of the experiment and the analyses outpaces this weakness. Furthermore, this limitation is transparently declared in the paper.

The other point regards the discussion about the results linked to low-CSE and high-CSE lower performance compared with average-CSE; I would like to suggest you to briefly improve it. I think that this evidence should be briefly commented citing the literature about decision-making style and performance. As a matter of fact, low-CSE and high-CSE are both groups of intuitive students, and the best performing group is the average-CSE that is the group more reflective. Therefore, in the discussion I would suggest you add some part about the link between decision-making styles (intuitive/analytical) and performance; for some references, you could check Bullni Orlandi and Pierce (2020), Elbanna et al. (2013), and Goll and Rasheed (1997).

To conclude, I think that the paper is worthy of being published in Administrative Sciences after the minor revision suggested.

Thank you again for giving me the chance to read your paper, and I hope my suggestions can be helpful.

References for the review

Bullini Orlandi, L., & Pierce, P. (2020). Analysis or intuition? Reframing the decision-making styles debate in technological settings. Management Decision.

Elbanna, S., Child, J., & Dayan, M. (2013). A model of antecedents and consequences of intuition in strategic decision-making: Evidence from Egypt. Long Range Planning46(1-2), 149-176.

Goll, I. and Rasheed, A.M.A. (1997). Rational decision-making and firm performance: the moderating role of environment, Strategic Management Journal, Vol. 18 No. 7, pp. 583-591.

Author Response

Ms. Ref. No.: admsci-784794

The Influence of Core Self-Evaluations on Group Decision Making Processes: A Laboratory Experiment

for

Administrative Sciences

Replies to reviewers

Dear Editor and Reviewers,

We would like to express our gratitude for your comments in this round of review. We thank the editors and reviewers for the guidance to refine our work.

Please find attached the revised version of our manuscript entitled “The Influence of Core Self-Evaluations on Group Decision Making Processes: A Laboratory Experiment”.

Thanks for reading and taking into consideration our manuscript. Your comments were well-received and the relative suggestions have been incorporated into this improved version. Your work truly enables us to significantly improve our own.

In the following section, we have reported the comments and our replies. In the text, all of the changes are highlighted in yellow.

***

Reviewer: 2

Recommendation: Minor revisions

General comment

Dear Authors,

Thank you very much for giving me the chance to read and review your paper about Core-Self Evaluation (CSE) and its relationships with intuitive/analytical information processing and decision-making performance. The topic is relevant and interesting in the actual debate.

The paper is really well-developed and written; I have only a couple of comments I hope can be helpful to improve the contribution of the paper.

Reply to general comment: Thank you for having appreciated the contribution, its aim, organization and development! This is a very great recognition for a scientific work and scholars behind it. Thank you, sincerely. We have improved the manuscript according to your suggested following comments.

Comment 1: Probably one of the main weaknesses of paper is the employment of graduate students as a unique source of data; nevertheless the rigor of the experiment and the analyses outpaces this weakness. Furthermore, this limitation is transparently declared in the paper.

Reply to Comment 1: You are more than right in saying that the main limitation of the paper is having involved a sample of students. This, of course, limits us in immediately generalizing the implications of this paper to practitioners. This point has been raised also by reviewer 3. Despite we cannot add any practitioners’ sample right now, we better detailed our approach in sampling students and given support to it. In particular, it has been added in the methodology that (lines 301-315 page 6):

“The selection of participants followed the convenience sampling approach (Given 2008), which is non-probability sampling that consists of selecting a sample from a part of the population that is close at hand. Despite the fact that a stream of scholars believe that students’ samples are not suitable for behavioral research aiming to provide working implications for practitioners (e.g., Gallander et al. 2001), another equally important stream of research (e.g., Lucas 2003; Thomas 2011) believes that these samples are appropriate in cases of research emphasis on basic psychological processes. Concerning the latter, according to Berkowitz and Donnerstein (1982): “the meaning the subjects assign to the situation they are in and the behavior they are carrying out plays a greater part in determining the generalizability of an experiment’s outcome than does the sample's demographic representativeness” (p. 249). So, here it is strongly believed that due to the aim of this research (thus, finding connections among CSE, overconfidence and decision-making performance) and the settings of the laboratory experiment, at least the internal validity of the research is guaranteed. This follows similar contributions on management decisions (see Cristofaro 2016) that consider students’ samples and investigate psychological variables against decision-making performance”.

Moreover, as suggested by Reviewer 3, we also add the specification that this target group can be considered as composed by so-called ‘Millennials’. So, within the discussion and implications we also added that this results, because emerging on young graduate students, can be applied to young practitioners that already work in organizations.

Comment 2: The other point regards the discussion about the results linked to low-CSE and high-CSE lower performance compared with average-CSE; I would like to suggest you to briefly improve it. I think that this evidence should be briefly commented citing the literature about decision-making style and performance. As a matter of fact, low-CSE and high-CSE are both groups of intuitive students, and the best performing group is the average-CSE that is the group more reflective. Therefore, in the discussion I would suggest you add some part about the link between decision-making styles (intuitive/analytical) and performance; for some references, you could check Bullni Orlandi and Pierce (2020), Elbanna et al. (2013), and Goll and Rasheed (1997).

Suggested references:

Bullini Orlandi, L., & Pierce, P. (2020). Analysis or intuition? Reframing the decision-making styles debate in technological settings. Management Decision.

Elbanna, S., Child, J., & Dayan, M. (2013). A model of antecedents and consequences of intuition in strategic decision-making: Evidence from Egypt. Long Range Planning, 46(1-2), 149-176.

Goll, I. and Rasheed, A.M.A. (1997). Rational decision-making and firm performance: the moderating role of environment, Strategic Management Journal, Vol. 18 No. 7, pp. 583-591.

Reply to Comment 2: Thank you for this comment. We agree with you on the fact that the relationship between intuition/reflective thinking and performance was not well developed in our work. We also thank you for the references you provided, they were very useful for developing this point. As to answer to your comment, we added the following part within the ‘Discussion and implications’ section (lines 447 to 466 of page 10):

“By linking the results of the test of the first and third hypothesis, it emerges that groups with a high level of intuitive thinking (corresponding to the ones having high or low CSE scores) are not the best performers in decision making terms. This result helps shedding light on the consequences of intuition in management decision making, an important avenue of research raised by Elbanna et al. (2013). In particular, it supports ‘a stream’ of prior results highlighting that intuitive thinking leads to poor quality of decisions (Elbanna et al, 2013), that consequently lead to poor firms’ performance (Goll and Rasheed, 1997). This happens, as suggested by the cited scholars, because intuitive decision makers are impatient with routine or details – they have a poor systemic search for information – and are pushed, by their nature, to quickly reach conclusions and have the consequence to ignore negative problems (also called ‘decision disturbance’). However, as well demonstrated by Bullini Orlandi and Pierce (2020), there is another important stream of research that has found a positive relationship of intuition with firm’s performance, such as in developing technologies, sizing new opportunities, and providing effective responses to crisis. What is determinant for the success of intuitive thinking seems to be, according to Bullini Orlandi and Pierce (2020), the dynamicity of the industry environment; indeed, in cases of highly dynamic and turbulent environments with presence of real-time data, analytical information processing is preferred rather than the reflective one”.

Finally, thank you for the comment on improving the communication style. We have asked Mrs. Alex Kessler, who has been a professional proofreader of academic works for the last 10 years and that is a native speaker of English, to assist us in polishing the work.

***

Finally, we again thank the Editor and Reviewers for their time, which has resulted in highly thoughtful reviews and invaluable guidance for the authors. The paper has evolved – structurally and in terms of content – and we anticipate that the improvements will meet expectations.

We hope that the revisions in the manuscript, and our accompanying responses, will be sufficient to make our manuscript suitable for publication in Administrative Sciences.

We shall look forward to hearing from you at your earliest convenience.

Yours sincerely,

The Authors

Reviewer 3 Report

Kindly find the attachment.

Author Response

Ms. Ref. No.: admsci-784794

The Influence of Core Self-Evaluations on Group Decision Making Processes: A Laboratory Experiment

for

Administrative Sciences

Replies to reviewers

Dear Editor and Reviewers,

We would like to express our gratitude for your comments in this round of review. We thank the editors and reviewers for the guidance to refine our work.

Please find attached the revised version of our manuscript entitled “The Influence of Core Self-Evaluations on Group Decision Making Processes: A Laboratory Experiment”.

Thanks for reading and taking into consideration our manuscript. Your comments were well-received and the relative suggestions have been incorporated into this improved version. Your work truly enables us to significantly improve our own.

In the following section, we have reported the comments and our replies. In the text, all of the changes are highlighted in yellow.

***

Reviewer: 3

Recommendation: Minor revisions

General comment

Thank you very much for submitting the manuscript titled “The Influence of Core Self- Evaluations on Group Decision Making Processes: A Laboratory Experiment”.

The topic of the article is exciting and modern. The authors refer to a decision-making problem that is poorly recognizing in the literature. The authors' work is important because it bridges the gap in empirical research. The theory is extensive in this area of study. The authors' research allowed the verification of the theory results in practice.

The authors' work is important because it bridges the gap in empirical research. The theory is extensive in this area of study. The authors' research allowed the verification of the theory results in practice.

The strength of the research is the combination of various fields of science. Research combining management and psychology is needed to develop employees and managers.

Reply to general comment: Thank you for having appreciated the contribution, its aim, organization and development.

Additional Questions

Comment 1: The title is understandable. It reflects well the purpose and content of the article.

Reply to Comment 1: Thank you for having appreciated the title.

Comment 2: It is worth that the authors highlight the purpose of the research and research question in the abstract.

Reply to Comment 2: Thank you for this comment. We implemented the abstract adding the following sentences (lines 9-13 of page 1): “This gap can be substantiated by the following research question: “How do high Core Self-Evaluations influence team decision-making processes?”. Answering it provides insights on how the evaluations that decision makers make about situations (and the consequent actions that are implemented) highly depend on decision makers’ inner traits and their effect on cognition”.

Comment 3: There is no justification for choosing a sample and an accurate description of the sample. Were these students from one university? Has students' professional experience been verified?

Reply to Comment 3: Thank you for this comment, we recognize that some important elements were not accounted. We added the following sentence within the methodology as to better describe the sample (lines 292-293 of page 16): students “following an optional management course in one large Italian University were involved in this research”. Professional experience has not been controlled as well as for other variables; we reported it as a limitation for this study. However, it is true that we did not mention work experience and we included it in page 11. Within the method, it has been also stated that: “Sampled students, as gleaned from informal conversation, had no or limited work experience; however, we did not control work experience in an empirical way”.

Comment 4: Answers to questions from Appendix A (Core-Self Evaluation Scale by Judge et al. 2003) require work experience and are focused on mindfulness.

Reply to Comment 4: Thank you for this comment. If you look at the works of Judge et al. (2003) and the one of Judge et al. (1998) you can spot that the validation of the questionnaire has been done on an extensive sample of students in the psychology field. In particular, in Judge et al. (2003) have been involved two samples of students counting 356 graduates in total, whose average age was 20.1 and 21.3. Moreover, Judge et al. (2003) state, in their work, that the reported questionnaire can be applied to everyone; this wants to be a trait scale as the MBTI or the Five Factor Model are, thus a questionnaire that can be filled by everyone.

Comment 5: It is challenging to formulate guidelines for managers based on a study conducted with the participation of students. Students do not have much professional experience. We can assume that they have no experience as a manager. It is more appropriate to indicate "Millennials" as participants in the study. Then you can refer the study results to this specific professional group.

Reply to Comment 5: You are more than right in saying that the main limitation of the paper is having involved a sample of students. This, of course, limits us in immediately generalizing the implications of this paper to practitioners. This point has been raised also by reviewer 2. Despite we cannot add any practitioners’ sample right now, we better detailed our approach in sampling students and given support to it. In particular, it has been added in the methodology that (lines 296-310 of page 6 and 7):

“The selection of participants followed the convenience sampling approach (Given 2008), which is non-probability sampling that consists of selecting a sample from a part of the population that is close at hand. Despite the fact that a stream of scholars believe that students’ samples are not suitable for behavioral research aiming to provide working implications for practitioners (e.g., Gallander et al. 2001), another equally important stream of research (e.g., Lucas 2003; Thomas 2011) believes that these samples are appropriate in cases of research emphasis on basic psychological processes. Concerning the latter, according to Berkowitz and Donnerstein (1982): “the meaning the subjects assign to the situation they are in and the behavior they are carrying out plays a greater part in determining the generalizability of an experiment’s outcome than does the sample's demographic representativeness” (p. 249). So, here it is strongly believed that due to the aim of this research (thus, finding connections among CSE, overconfidence and decision-making performance) and the settings of the laboratory experiment, at least the internal validity of the research is guaranteed. This follows similar contributions on management decisions (see Cristofaro 2016) that consider students’ samples and investigate psychological variables against decision-making performance”.

Moreover, we already highlighted that involving students instead of managers is a huge limit of this study. We extensively report it in page 11.

About the word ‘Millennials’, we inserted it within the methodology (line 293 of page 16) and implications for practitioners when referring to the involved students; we are really grateful to you for the intuition of extending the generalizability of results to these ‘young practitioners’.

Comment 6: The authors should complete the description of the research method used. They did not substantiate why chosen the experiment. The experiment has not been characterized in detail. If one of the researchers wanted to repeat the experiment, there is not enough data and information.

Reply to Comment 6: Thank you for this comment. We fixed the gaps you highlighted. In particular, we added the following point as to detail why we chose a laboratory experiment as research design (lines 294-298 of page 6):

“The laboratory experiment, then followed by quantitative analysis, is considered as the most suitable research design in research fields that can be considered at a stage of development far from the nascent one (Edmondson and McManus, 2007). These intermediate or mature fields of research – like the role of personal traits in decision making processes (see the review by Cristofaro, 2017a) – are challenged by “focused questions and/or hypothesis relating existing constructs” (p. 1160)”.

We added some details in the description of the laboratory experiment (lines 315-325 of page 7). We really think that, right now, whoever researcher has in her/his hands the info to replicate the experiment. It is worth notice that other reviewers found details about the implemented methodology very satisfactory.

Commnent 7: The authors should indicate directions for future research.

Reply to Comment 7: Thank you for this comment. You are right in saying that future research directions were not well exploited. At page 12 we have highlighted all the future research that can be taken from this work. We have firstly pointed out that other scholars can overcome the many limits of this paper; then, they can also look at the work of Abatecola et al. (2018) and finding whether there could be other biases linked with the CSE, not only overconfidence. Finally, it has been added in this version of the manuscript that (lines 510-518):

“Yet, the results of this work should be highly considered by scholars that want to deepen the antecedents of intuition and its outcomes in strategic decision-making (see Elbanna et al. 2013). In particular, future research can test whether trait variables –CSE above all – have more weight than contextual and environmental ones in determining the thinking style of decision makers. Last but not least, the link between CSE and performance could be deepened also by looking at the emotional answers that high, average, and low CSE groups have when facing some decisional situations. These can reinforce the debate and operationalization of intuitive thinking which, nowadays, still does not take the role of emotions in substantiating intuitive answers into very high consideration”.

Finally, thank you for the comment on improving the communication style. We have asked Mrs. Alex Kessler, who has been a professional proofreader of academic works for the last 10 years and that is a native speaker of English, to assist us in polishing the work.

***

Finally, we again thank the Editor and Reviewers for their time, which has resulted in highly thoughtful reviews and invaluable guidance for the authors. The paper has evolved – structurally and in terms of content – and we anticipate that the improvements will meet and exceed expectations.

We hope that the revisions in the manuscript, and our accompanying responses, will be sufficient to make our manuscript suitable for publication in Administrative Sciences.

We shall look forward to hearing from you at your earliest convenience.

Yours sincerely,

The Authors